# Genome-Wide Identification and Analysis of the *WRKY* Gene Family and Cold Stress Response in *Acer truncatum*

**DOI:** 10.3390/genes12121867

**Published:** 2021-11-24

**Authors:** Yan Li, Xiang Li, Jiatong Wei, Kewei Cai, Hongzhi Zhang, Lili Ge, Zengjun Ren, Chunli Zhao, Xiyang Zhao

**Affiliations:** 1College of Forestry and Grassland, Jilin Agricultural University, Changchun 130118, China; ly2019@nefu.edu.cn; 2State Key Laboratory of Tree Genetics and Breeding, School of Forestry, Northeast Forestry University, Harbin 150040, China; lx2019@nefu.edu.cn (X.L.); wjt2019@nefu.edu.cn (J.W.); ckw1006@nefu.edu.cn (K.C.); 3Linjiang Forestry Bureau of Jilin Provenance, Linjiang 134600, China; zhang11230805@163.com (H.Z.); pipiping123456@163.com (L.G.); 4Wangqing Forestry Bureau Shajingou Forestry Station of Jilin Provenance, Wangqing 133200, China; zhaoqs1002@163.com

**Keywords:** *Acer truncatum*, genome-wide, *WRKY* transcription factors, bioinformatics analysis, gene expression

## Abstract

*WRKY* transcription factors constitute one of the largest gene families in plants and are involved in many biological processes, including growth and development, physiological metabolism, and the stress response. In earlier studies, the *WRKY* gene family of proteins has been extensively studied and analyzed in many plant species. However, information on *WRKY* transcription factors in *Acer truncatum* has not been reported. In this study, we conducted genome-wide identification and analysis of the *WRKY* gene family in *A. truncatum*, 54 *WRKY* genes were unevenly located on all 13 chromosomes of *A. truncatum*, the highest number was found in chromosomes 5. Phylogenetic relationships, gene structure, and conserved motif identification were constructed, and the results affirmed 54 *AtruWRKY* genes were divided into nine subgroup groups. Tissue species analysis of *AtruWRKY* genes revealed which were differently exhibited upregulation in flower, leaf, root, seed and stem, and the upregulation number were 23, 14, 34, 18, and 8, respectively. In addition, the *WRKY* genes expression in leaf under cold stress showed that more genes were significantly expressed under 0, 6 and 12 h cold stress. The results of this study provide a new insight the regulatory function of *WRKY* genes under abiotic and biotic stresses.

## 1. Introduction

Transcription factors are the most abundant gene regulators in multicellular genomes. They activate or inhibit the expression of target genes by binding to specific DNA sequences, thus regulating the gene expression of all organisms [1,2,3]. *WRKY* transcription factors are one of the largest and most important families of transcription factors with a highly conserved protein structure domain [4,5]. Structurally, approximately 60 amino acid residues of the N-terminus contain the conserved sequence associated with DNA binding activity (WRKYGQK), and the C-end have a zinc-finger motif (Cx_4–5_Cx_22–23_HxH or Cx_7_Cx_23_HxC) to participate in zinc finger protein interactions [6,7,8]. WRKY proteins are divided into three groups based on the number of conserved domain and type of zinc finger, named group Ⅰ, group Ⅱ and group Ⅲ [9,10]. Group Ⅰ comprises two Cx_4–5_Cx_22–23_HxH zinc figure motifs; group Ⅱ contains one Cx_4–5_Cx_22–23_HxH zinc finger motif; and group Ⅲ includes one Cx_7_Cx_23_HxC zinc finger motif [6]. On the basis of their phylogenetic clades and different assembling, the three groups can be further divided into subgroups, such as group Ⅱ have five subgroups (Ⅱ a–e) [11,12]. WRKY has many special biological functions due to its unique domain.

Numerous studies have indicated the crucial regulatory roles of *WRKY* transcription factors in plant growth and development, physiological and biochemistry processes, biotic and abiotic stress response [13,14,15,16]. For instance, overexpression of the transcription factor *TaWRKY2* enhances drought stress tolerance and increase grain yield in transgenic wheat (*Triticum aestivum* L.) [17]; while overexpression of *OsWRKY29* represses seed dormancy by directly downregulating the expression of *OsABF1* and *OsVP1* in rice (*Oryza sativa* L.) [18]. What is noteworthy is that the regulatory functions of *WRKY* genes are closely associated with multiple plant hormone-mediated signal pathways. In *Arabidopsis thaliana*, *AtWRKY46* modulated the development of lateral roots in osmotic/salt stress conditions via regulation of ABA signaling and auxin homeostasis [4]. Cold tolerance is a response of plants to abiotic stress conditions, the gene expression levels of plants will change differently under cold stress. Many studies have proved that members of the *WRKY* family play a crucial regulatory role in cold stress, for instance, the expression of some *PmWRKY* genes was induced by cold stress in *Prunus mume*, which show 6 *PmWRKY* genes were downregulated, and three genes were upregulated expressed sustainably with prolonging of the treatment time in stems [19]. In *Camellia sinensis*, *CsWRKY33*, *CsWRKY34*, *CsWRKY37*, *CsWRKY38* and *CsWRKY39* genes were upregulated under low (4 °C) temperature treatments [20].

*Acer truncatum*, a deciduous tree belonging to the Aceraceae, is a crucial landscaping tree species with high ornamental and economic value [21,22]. The natural distribution range of *A. truncatum* is mainly concentrated in northern China, Korea and Japan, but there are a few sporadically distributed species in Europe and North America [23,24]. Due to its elite hardwood, it is widely used for timber production. Seeds and leaves are rich in oils, nervonic acid and tannins, which can also be used as raw materials for food processing and pharmaceutical development [25,26,27]. Therefore, increasing attention has been given to the various uses of *A. truncatum*, such as ornamental greening, medicinal value and ecological benefit, and research on *A. truncatum* has also increased. In recent years, research on the molecular biology aspects of *A. truncatum* has mainly focused on molecular markers, genetic diversity, and drug synthesis, but studies on its gene family are still lacking. Publication of the whole genome sequence of *A. truncatum* makes up for the gap in molecular biology research [28], which will be of great significance for further studies on *A. truncatum* flowering, seed production, and biological and abiotic stress.

In this study, we comprehensively analyzed the *WRKY* gene family with multiple bioinformatics methods and further determined the function of *WRKY* in growth and development. Furthermore, some *AtruWRKYs* were preliminarily verified in regulating *A. truncatum* tolerance to cold stress, and the transcript level of these responsive *WRKY* genes influence *A. truncatum* response to cold stresses. The results lay a theoretical foundation for deeper research on *WRKY* genes.

## 2. Materials and Methods

### 2.1. Plant Materials

The 3-year-old *A. truncatum* used in this study from the Northeast Forestry University greenhouse (126°38′8.92″ E, 45°43′20.64″ N), Harbin, Heilongjiang province, China. The *A. truncatum* seedlings was positioned in a low-temperature refrigerator at 4 °C to experimentally validate the computationally predicted *AtruWRKY* genes, and leaves of samples were collected at 6, 12, 24, 36 and 48 h after treatment with 0 h as a control. The collected leaves were instantly frozen in liquid nitrogen for 5 min and stored at −80 °C until they were used for extracting total RNA.

### 2.2. Sequence Retrieval of the WRKY Gene Family in A. truncatum

The *A. truncatum* genome files v1.1 (such as *A._truncatum*.gff, *A._truncatum*.pep and *A._truncatum*_genome.fa) was retrieved in https://doi.org/10.6084/m9.figshare.12986237.v2, accessed on 8 July 2021 and used to identify *AtruWRKY* [28]. Next, 74 known *WRKY* transcription factor family genes from *A. thaliana* were selected as the query objects, and we obtained the Arabidopsis protein sequence by TAIR 9.0 release (https://www.arabidopsis.org/browse/genefamily/index.jsp, accessed on 10 July 2021) [29]. The protein sequences of *A. truncatum* and *A. thaliana* were subjected to BLAST alignment by TBtools (Toolbox for Biologists) v1.086 (evalue <1 × 10^−5^) [30], each *A. thaliana* gene was successfully matched with multiple *AtruWRKY* genes (Appendix A). Further, a total of 572 alignment sequence IDs were obtained after eliminating the repetition values and blanks. Then, candidate protein sequences were extracted by TBtools. In addition, the candidate *AtruWRKY* and *A. truncatum* were further manually analyzed using Batch CD-Search in the National Centre for Biotechnology database (NCBI; https://www.ncbi.nlm.nih.gov/Structure/bwrpsb/bwrpsb.cgi, accessed on 11 July 2021) to confirm the presence of the WRKY domain. We estimated 7 conserved domains in *A. truncatum*: WRKY, UPF0242 superfamily, Plant zinc cluster domain (Plant_zn_clust, PF10533), PLN0321 superfamily, PKc-like superfamily, and PAH^−^ and Sin3 superfamily. Finally, only the genes containing conserved WRKY domains were selected for subsequent analysis, and a total of 54 *AtruWRKYs* were obtained and termed *AtruWRKY1* to *AtruWRKY54* (Table 1).

The proteomics and sequence analysis tools on the ExPASy (http://expasy.org/, accessed on 13 July 2021) proteomics server was used to predict the protein molecular weights (MW) and isoelectric points (PI) of *AtruWRKY* genes. To ensure the accuracy and completeness of the data, the CDS length and amino acid sequence were predicted by TBtools [31]. In addition, subcellular localization (https://wolfpsort.hgc.jp/, accessed on 13 July 2021) were calculated through the online website.

### 2.3. The Gene Structure and Chromosomal Location

To predict the exon-intron structure, the gene structure of all candidate *A. truncatum WRKY* genes was identified by TBtools (used *A._truncatum*.gff and genes ID), and TBtools software was used for visualization (from GTF/GFF3 File) [32,33]. At the same time, TBtools was used to determine the chromosomal location of the *AtruWRKY* genes, and their gene ID and sequences were used as the basic data for the searches. According to the gene location and number, each *AtruWRKY* gene was mapped to *A. truncatum* chromosomes, where tandemly duplicated gene pairs were linked with a red line [34,35]. It is well known that *A. truncatum* has 13 pairs of chromosomes [28], so they were named chr1, chr2, chr3, chr4, chr5, chr6, chr7, chr8, chr9, chr10, chr11, chr12 and chr13 in this study.

### 2.4. Protein Motif Composition Analysis and Functional Annotation

To identify the conserved motifs in *A. truncatum* proteins, the motifs within the 54 *A. truncatum*
*WRKY* protein sequences were detected using the Multiple EM for Motif Elicitation (MEME 5.3.2: https://meme-suite.org/meme/doc/meme.html?man_type=web, accessed on 20 July 2021) [36]. The maximum number of motifs was set to 10, and members of the same subfamily have similar conserved domain characteristics. In addition, GO annotation was performed by Eggnog (http://eggnog5.embl.de/, accessed on 20 July 2021) and visualized using WEGO 2.0 (https://wego.genomics.cn/, accessed on 21 July 2021).

### 2.5. Sequence Alignment and Phylogenetic Tree Construction

According to the conserved domain of the *A. truncatum*
*WRKY* protein, the family can be divided into different groups. The Clustalw algorithm analysis was performed, and the multiple sequence alignment analysis using the default parameters in MEGA6.0: 1000 replicates for bootstrap analysis and pairwise deletion [37,38]. Then, the phylogenetic tree was constructed from the alignment results using neighbor-joining (NJ) method. The phylogenetic tree was annotated and beautified by using iTOL online software (https://itol.embl.de/, accessed on 24 July 2021) [39,40]. Additionally, we downloaded the genomic information of *A. thaliana* (TAIR10.1; https://www.ncbi.nlm.nih.gov/genome/?term=+Arabidopsis+thaliana, accessed on 8 July 2021) and a related species called *Acer yangbiense* (AYv1.1; https://www.ncbi.nlm.nih.gov/genome/?term=Acer+yangbiense+, accessed on 29 July 2021) from the NCBI website and analyzed the collinearity with the information for the three species using TBtools [41].

### 2.6. Analysis of AtruWRKY Gene Expression in Different Tissues/Organs

To analyze the expression profile of *AtruWRKY* genes in different tissues/organs, RNA-seq data of *AtruWRKY* genes were downloaded from the NCBI databases (https://www.ncbi.nlm.nih.gov/sra, accessed on 3 September 2021), which contains the expression levels in root (SRR10097461), leaf (SRR10097462), flower (SRR10097463), stem (SRR10097460) and seed (SRR10097465). The expression abundance of *AtruWRKY* genes were calculated using the fragments per kilobase of transcript per million fragments mapped (FPKM) values. R studio (pheatmap; https://cran.r-project.org/web/packages/pheatmap/index.html, accessed on 4 September 2021) was used for standardized analysis and visualization of gene expression data.

### 2.7. Real-Time qRT-PCR Experimental Validation

The OMEGA Biotek (Guangzhou Feiyang Biological Engineering Co., Ltd., Guangzhou, China) was used to extract total RNA from leaves (≤100 mg), and the experimental operations were carried out strictly according to the instructions (Version: Plant RNA Kit R6827; http://omegabiotek.com.cn/template/productShow.aspx?m=129002&i=100000111413546, accessed on 10 September 2021). In addition, the RNA integrity and RNA purity (OD260/230 and OD260/280) of the samples were determined by agarose gel electrophoresis and ultrasonic spectrophotometry. To obtain the the amplification products of cDNA for qRT-PCR analysis, approximately 1 ug of total RNA was reverse-transcribed using the PrimeScript RT reagent kit with gDNA Eraser (TaKaRa, Kyoto, Japan). The primers were designed using the online website INTEGRATED DNA TECHNOLOGIES (https://sg.idtdna.com/pages, accessed on 22 September 2021), a total of 15 *WRKY* genes were selected randomly and used for RT-qPCR analysis with specific primers, and 18S rRNA was used as a reference gene (Appendix A). The amplicon size ranging from 150 bp to 200 bp and an optimal Tm of 62 ± 1 °C. Quantitative real-time qRT-PCR was performed on an ABI 7500 Real-Time system (Applied Biosystems) using the TaKaRa SYBR Green Mix kit (TaKaRa, Beijing, China). The PCR protocol was conducted with a 20 µL volume, which contained 0.4 uL of ROX Reference Dye Ⅱ, 0.8 uL of upstream and downstream primers (10 umol/L), 2 uL of cDNA template, 6 uL of double-distilled water (ddH_2_O) and 10 uL of 2 × SYBR (TB Green Premix Ex Taq Ⅱ). The PCR reaction program consisting of 95 °C for 30 s, followed by 40 cycles of 95 °C for 5 s and 62 °C for 35 s, 95 °C for 15 s, and 60 °C for 1 min, finishing with 95 °C for 15 s. Three technical repetitions were performed for the whole experiment, the relative expression level was calculated using the 2^−^^ΔΔ^^CT^ method.

## 3. Results

### 3.1. Identification of AtruWRKY in A. truncatum

Based on the amino acid sequences of the *A. thaliana* WRKY gene family, a total of 54 *AtruWRKY* genes were successfully identified from the *A. truncatum* genome by multiple sequence alignment after removing duplicates, incomplete sequences, and sequences without corresponding domains, and they were named *AtruWRKY1* to *AtruWRKY54* according to their Gene ID and structure. Detailed information on the 54 *WRKY* genes is provided in Table 1. The number of amino acids (aa) in the predicted protein varied from 71 to 768. The average predicted isoelectric points and molecular weight points of the encoded proteins were 7.01 and 42,388.03, respectively. The subcellular localization results showed that most of the *WRKY* genes (94%) were localized in the nucleus, and only the *AtruWRKY1*, *AtruWRKY5* and *AtruWRKY24* genes were localized in the cytosol.

### 3.2. Phylogenetic Tree Construction and Conserved Motifs

To further explore the phylogenetic relationship of the *WRKY* transcription factor family in *A. truncatum*, a phylogenetic tree of *A. truncatum* was constructed by using MEGA software as shown in Figure 1, which intuitively reflects the evolutionary status and grouping attribution of 54 members of the *WRKY* family. According to 54 *WRKY* protein sequences and clustering analysis, the 54 identified members of the *WRKY* family were sorted into three large groups. Specifically, the largest number of *WRKY* members in the third largest group was 29. The first and second groups had 24 and 11 members, respectively. Furthermore, a total of 10 motifs were identified by analyzing the conserved motifs of *WRKY* family members through online MEME. Based on the number and amino acid sequences of the conserved *WRKY* domains, we divided the three large groups into nine subgroups (Ⅰ a, Ⅰ b, Ⅱ a, Ⅱ b, Ⅲ a, Ⅲ b, Ⅲ c, Ⅲ d and Ⅲ e). As shown in Figure 2, the same subgroups of WRKY members usually had similar motifs. For instance, almost all gene had more than 4 motifs, and all genes except *AtruWRKY5* and *AtruWRKY31* contained motif 2, which might play a crucial role in *A. truncatum* proteins. Interestingly, motif 5 was observed only in group Ⅰ b, whereas motif 9 was only found in groups Ⅲ d and Ⅲ e. The group Ⅰ b had the highest average number of motifs, approximately 8.

### 3.3. Chromosomal Location and Gene Structure

All 54 *AtruWRKY* genes were mapped to 13 chromosomes (2n = 2X = 26) in the *A. truncatum* genome (Figure 3). Notably, although *WRKY* genes were not evenly distributed on chromosomes, they were present on every chromosome. Chr5 contained the most *AtruWRKY* genes (9), the number of genes located on other chromosomes ranged from 1 to 6, and the lowest number of genes appeared on chr13 (1). There were six *AtruWRKYs* on chr6 and chr8 and only two *AtruWRKYs* on chr1 and chr3. Gene tandem duplication is considered to be an important reason for the formation of gene clusters [42,43]. In this study, we discovered that some *AtruWRKYs* were adjacent to each other. For example, *AtruWRKY14* and *AtruWRKY17* on chr7 were connected in series, suggesting that there may be a tandem duplication relationship between these *AtruWRKYs*. Additionally, we further investigated the *WRKY* exon-intron coding sequence structure to obtain in-depth insight into the protein expression sequence. The results showed that each group had a highly conserved structure (Figure 4). The distribution of CDS, UTR and introns of *WRKY* genes had a certain regularity, the intron number in the *A. truncatum*
*WRKY* family ranged from 1 (*AtruWRKY46*) to 5 (*AtruWRKY21*), with an average number of 2.74, and group Ⅰ genes contained more introns. The number of CDSs was evenly distributed in *AtruWRKY* genes, ranging from 3 to 4, but there was no UTR in many *AtruWRKY* (5, 6, 8, 11, 14, 17, 18, 24, 27, 28, 30, 36, 39, 40, 43, 45, 49, 52) genes.

### 3.4. Synteny Analysis of AtruWRKY Genes

To reveal the origin and evolution of the *A. truncatum* WRKY family members, a synteny analysis was performed between *WRKY* genes in *A. truncatum* and two other plants, including *A. thaliana* and *A. yangbiense* (Figure 5). Red lines in the background highlight syntenic *WRKY* gene pairs within *A. truncatum* and other plant genomes, while gray lines indicate collinear blocks. Fifty and 65 collinear gene pair showed syntenic relationships with those in the other two species: *A. thaliana* and *A. yangbiense*, respectively. *A. truncatum* and *A. yangbiense* are both members of Aceraceae, and *WRKY* genes show stronger homology than *A. thaliana*.

### 3.5. AtruWRKY Expression Profiles in Five Tissues

The expression levels of all 54 *AtruWRKYs* were investigated thoroughly using a rigorous transcriptome analysis procedure based on public transcriptomic data of different tissues of *A. truncatum*, including flowers, leaves, roots, seeds and stems. Among the 54 *AtruWRKY* genes, 52 *AtruWRKYs* (except *AtruWRKY49* and *AtruWRKY53*) were identified in all tissue types (Figure 6). The *AtruWRKY* genes showed different expression across tissues tested; one gene in the flower (*AtruWRKY11*) exhibited the highest transcript levels, and the expression of *AtruWRKY34*, *AtruWRKY46* and *AtruWRKY5* occurred preferentially in seeds. Six *AtruWRKY* genes (*AtruWRKY24*, *AtruWRKY9*, *AtruWRKY8*, *AtruWRKY2*, *AtruWRKY18* and *AtruWRKY30*) showed higher expression levels in roots than in seeds and stems. The expression analysis of the different leaf developmental stages showed that several genes (*AtruWRKY52*, *AtruWRKY15* and *AtruWRKY17*) had higher expression in the stem.

### 3.6. Functional Annotation

Of all the *AtruWRKY* genes identified, most (47 genes) were successfully annotated for their functions (Figure 7). By analyzing the cellular components of the functionally annotated genes, we found that only a few genes played a part in membrane structure formation, such as organelle parts (one gene), membrane-enclosed lumen (one gene), extracellular region (one gene) and membrane (one gene). However, there are many genes that are widely present and have positive effects on molecular function (46 genes), accounting for more than 95% of all genes. We further analyzed the biological processes and found that there were significant differences in the expression of the number of genes, the number of genes annotated for each GO term can be found on the website (https://wego.genomics.cn/view/WEGOID77554007142815, accessed on 21 July 2021).

### 3.7. Expression Analysis of AtruWRKY Genes under Cold Stress

To verify the functions *WRKYs* play in cold hardiness, the expression patterns of 15 randomly selected *AtruWRKY* genes in different stages at low temperature (4 °C) (0, 6, 12, 24, 36 and 48 h) were determined by qRT-PCR. The results showed that 15 *AtruWRKY* genes exhibited significant differences, and most of the genes were highly expressed during 0~12 h under cold treatment (Figure 8). *AtruWRKYs*, including *AtruWRKY12*, *AtruWRKY13*, *AtruWRKY15*, *AtruWRKY17*, *AtruWRKY28*, *AtruWRKY31*, *AtruWRKY39*, *AtruWRKY44* and *AtruWRKY47*, showed the highest expression levels when exposed to low temperature for 12 h. *AtruWRKY29*, *AtruWRKY33*, and *AtruWRKY51* showed the highest levels at 6 h, however *AtruWRKY20* (48 h), *AtruWRKY25* (24 h) and *AtruWRKY51* (36 h) also showed relatively high expressed level after 12 h of cold treatment. Notably, the *AtruWRKY33* gene was only distinct expressed at 6 h, while *AtruWRKY15*, *AtruWRKY17* and *AtruWRKY33* were almost non-expressed at 12~48 h.

## 4. Discussion

*A. truncatum* is a well-known and valuable tree species for its graceful maple, elite hardwood and rich medicinal ingredients [44,45]. However, only a few studies have been conducted on its growth development and stress response at molecular level because of incomplete RNA-seq and genome sequence data. Previous studies have found that TFs play a crucial role in plant growth and response to abiotic stress through self-regulation and regulation of downstream target gene expression [46,47,48]. *WRKY* transcription factors have been reported to be one of the largest gene families and play a pivotal role in flowering, growth development, and abiotic and biotic stress responses in plants [16]. So far, some *WRKY* family members have been identified and analyzed in plant species such as *A. thalliana* [49], *G. max* L. [50], *T. aestivum* [17] and *O. sativa* [51]. However, none of the reports are published for the identification and functional role of the *WRKY* gene family in *A. truncatum*. This study is the first publicly published analysis and identification of *WRKY* transcription factors via genomic data, which provided a better understanding of the function of *WRKY* genes family under cold stress in *A. truncatum*.

The number of TFs in a gene family is related not only to the genetic background of the species but also to the influence of the long-term evolutionary succession of plants. In present investigation, 54 genes were ultimately identified as encoded by the *A. truncatum* WRKY family. The number of *WRKY* genes identified in *A. truncatum* was neither high nor less than those *WRKY* genes in various plants (*Santalum album*, 57 *WRKYs*; *Solanum tuberosum*, 79 *WRKYs*; *Sesamum indicum*, 71 *WRKYs*; *Panicum miliaceum* L., 32 *WRKYs*; *Zanthoxylum bungeanum* Maxim, 38*WRKYs*; *Taraxacum antungense* kitag, 44 *WRKYs*) [52,53,54,55,56,57], suggesting high conservation of WRKY gene family in *A. truncatum*, which may be related to gene duplication during species formation and evolution. Furthermore, the subcellular localization results in this study showed that all but *AtruWRKY1*, *AtruWRKY4* and *AtruWRKY24* were detected in the nucleus, suggesting that most *WRKY* gene functions may are closely related to the expression regulation of target genes, similar to *Xanthoceras sorbifolium* [58] and *S. album* [52]. The 54 *WRKY* transcription factors identified, all of which were unevenly distributed on 13 chromosomes in *A. truncatum*, may play important roles in the evolution of the *WRKY* gene family. Interestingly, the *AtruWRKY14* and *AtruWRKY17* have a tandem duplication relationship between them, which were also close phylogenetically. At the same time, transcription factor genes with similar functions were clustered together in evolutionary trees. In particular, group Ⅲ had significantly more gene members than the other two, accounting for 54% of all *AtruWRKYs*. This was consistent with the results of *P. miliaceum* L. [55] and Chinese jujube (*Ziziphus jujuba* Mill.) [8], indicating the *WRKY* gene family diversification and conservation among the land plants. In addition, some studies have clarified that the variation of group Ⅲ *WRKY* genes may be key to the variation of *WRKY* transcription factors [59].

The number and species of conserved motifs of each *WRKY* family member were different to some extent, but the conserved motifs and species of the protein members in the same subgroup were roughly the same, indicating that the members of the *WRKY* family had similar structure and biological function. All of *WRKY* genes family members in this paper contained motif 2, indicating that motif 2 with high frequency may be closely related to the molecular function and structural properties involved in *WRKY* gene. In-depth analysis of the structural characteristics of introns and exons is a key step in the process of gene family evolution [60,61]. A varied number of introns was possessed by *AtruWRKY* genes, group Ⅰ had more introns in this study, implying that the molecular structure in the group Ⅰ of *WRKY* genes may be quite conserved in the process of evolution, which conductive to the protein diversity caused evolution [62,63]. It seems that the crucial function of the *WRKY* gene family in *A. truncatum* is very much related to group Ⅰ, and similar results were found in eggplant (*Solanum torvum* L.) [60]. These results confirm the characteristics of the *AtruWRKY* gene family and facilitate further study on the function of *AtruWRKY* genes.

RNA-seq is usually used to study gene function and structure at the overall level and reveal the molecular mechanisms of specific biological processes and disease occurrence. This approach has been widely used in basic research, clinical diagnosis, drug development and other fields [64,65,66,67]. As shown in Figure 6, we used transcriptome data from different tissues/organs (root, flower, leaf, seed and stem) of *A. truncatum* to explore the expression of the *WRKY* gene family, the expression pattern of each *AtruWRKY* gene was altered in these tissues/organs. We found that among all 54 *AtruWRKY* genes, 52 genes (except *AtruWRKY49* and *AtruWRKY53*) were expressed specifically in plant tissues. Among them, most *AtruWRKY* genes were highly expressed in root (56%), whereas a few *AtruWRKY* genes were expressed in seed (6%) and stem (8%). This is consistent with studies made in other plants, such as *S. indicum* L. [54], cabbage (*Brassica rapa* ssp. pekinensis) [68], grape (*Vitis vinifera*) [69] and cucumber (*Cucumis sativus*) [70]. The results showed that the *WRKY* genes are expressed tissue specifically, and which may function reflect in responses that first affect plants below ground. *AtruWRKY2*, *AtruWRKY8*, *AtruWRKY9*, *AtruWRKY22* and *AtruWRKY24* showed the highest expression levels in the roots, and these genes may play a key role in the root formation and development of *A. truncatum*. For instance, previous studies have suggested that *TaWRKY51* genes took part in *T. aestivum* L. lateral roots formation by modulating ethylene biosynthesis [71]. Overexpression *AtWRKY75* and *OsWRKY31* gave rise to reduced significantly the number of relevant lateral roots in contrast [72,73]. *AtruWRKY11* is highly expressed only in flowers, indicating that some *AtruWRKY* might only be expressed in response to particular biotic and abiotic. These results suggest that *WRKY* transcription factors exhibit varied expression profiling in various organs or tissues to regulate various biological and physiological metabolism processes in *A. truncatum* [74,75].

Cold stress is a crucial factor affecting the growth and development of plants. Previous studies on the mechanism by which *WRKY* regulates cold stress have mainly focused on model plants, for instance, the *OsMADS57* and *OsTB1* conversely affect *O. sativa* L. chilling tolerance via targeting *OsWRKY94* [76]. In *A. thaliana*, the *AtWRKY34* negatively mediated cold sensitivity of mature pollen, speculated that it might be involved in the CBF signal cascade in mature pollen [77]. In this study, 15 differentially expressed *AtruWRKY* genes were verified in *A. truncatum* from the 54 genes (Figure 8). The results showed that 13 *AtruWRKY* genes were highly expressed at 0~12 h after low-temperature treatment, which implied that these genes may be crucial in the early stage of cold stress treatment. In contrast, the expression levels of *AtruWRKY25* and *AtruWRKY51* were slightly increased after 12 h, suggesting that these genes participated in the late reaction to cold treatment. This difference in expression before and after indicates that *WRKY* transcription factors were time-efficient in response to cold stress, which was also verified in *Brassica napus* [78] and *P. mumu* [19] in the past research, suggesting that different WRKY transcription factors may play roles in different periods. In addition, the expression of a few *AtruWRKY* genes decreased or faded away under cold treatment, such as *AtruWRKY17*, *AtruWRKY31*, *AtruWRKY33*, *AtruWRKY39* and *AtruWRKY47*. We conjecture that these genes may play a role in other biotic and abiotic stresses in *A. truncatum*.

## 5. Conclusions

This study is the first genome-level description of the *WRKY* gene family of *A. truncatum*. We identified 54 *WRKY* genes in *A. truncatum*, and all of them were located on 13 chromosomes. In addition, we identified ten conserved domains of *AtruWRKY* proteins, and these *WRKY* genes were classified into 3 groups (9 subgroups) based on phylogenetic relationships. The collinearity of *A. truncatum* and *A. yangbiense* was better than that of *A. thaliana*, and the functional annotation results showed that the majority of *WRKY* genes were involved in the regulation of molecular functions and biological processes, such as transcription regulator activity, biological regulation, regulation of biological process, metabolic process, and cellular process, etc. Furthermore, the expression patterns in five different tissues suggested that these *WRKY* genes might play a crucial role in flowers, seeds, leaves, roots, and stems. Finally, more *AtruWRKY* genes were significantly highly expressed under 0, 6 and 12 h of cold stress, which provides a meaningful direction for future research under cold stress. The analysis of *WRKY* genes identifies their molecular mechanisms and potential functions involved in plant biotic and abiotic stress responses in *A. truncatum* and lays a foundation for the study of WRKY TFs in *A. truncatum* and other plants.

## Figures and Tables

**Figure 1 genes-12-01867-f001:**
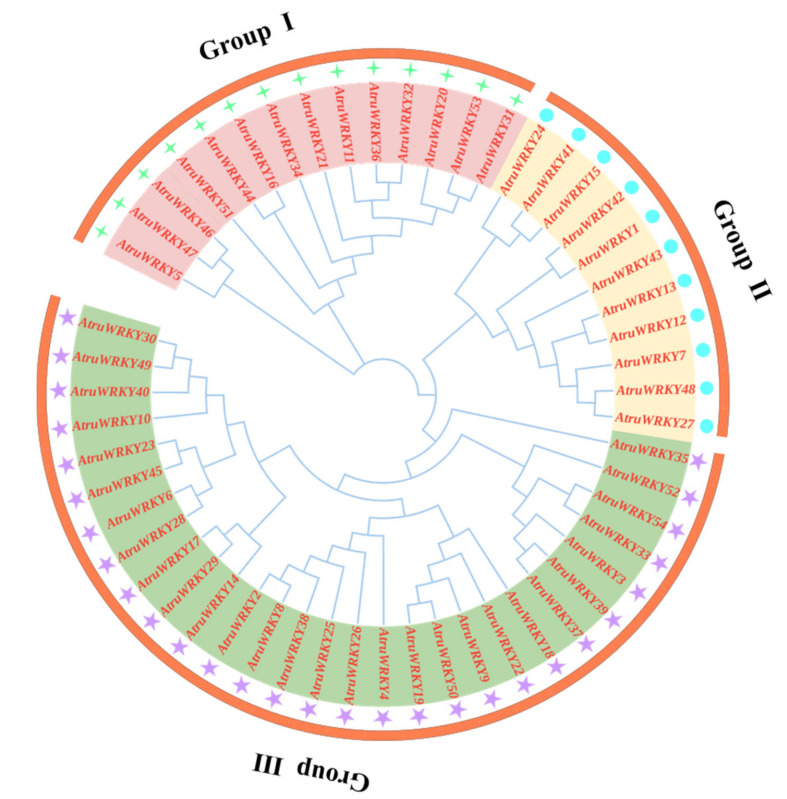
Phylogenetic tree of the *AtruWRKY* proteins from *Acer truncatum*. The NJ tree was constructed from the amino acid sequences of *AtruWRKY* using MEGA6.0 with 1000 bootstrap replicates. The *A. truncatum* WRKY proteins were grouped into three groups (Ⅰ, Ⅱ, Ⅲ), which were decorated with different colors.

**Figure 2 genes-12-01867-f002:**
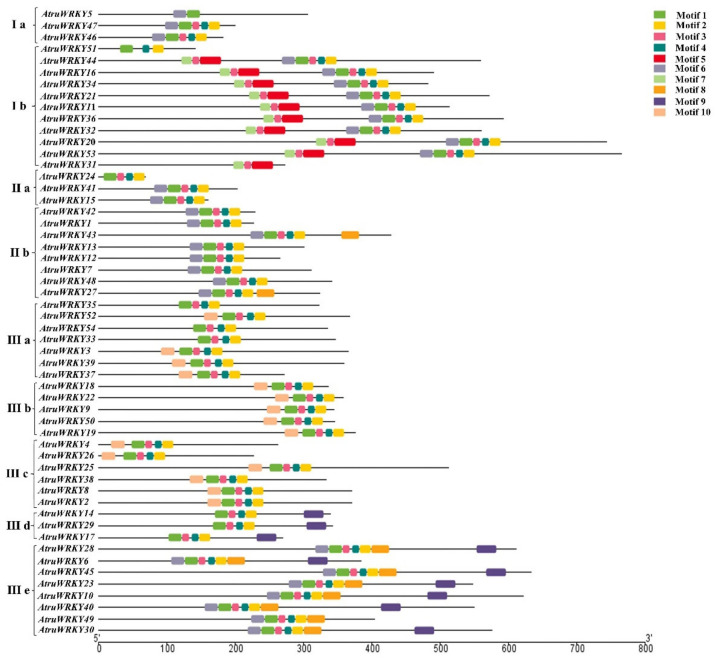
Conserved motifs of the proteins arranged according to their phylogenetic relationships with nine subgroups. The motifs in the *AtruWRKY* were identified using MEME5.3.2 online program version. The 10 conserved motifs are shown in diverse colors.

**Figure 3 genes-12-01867-f003:**
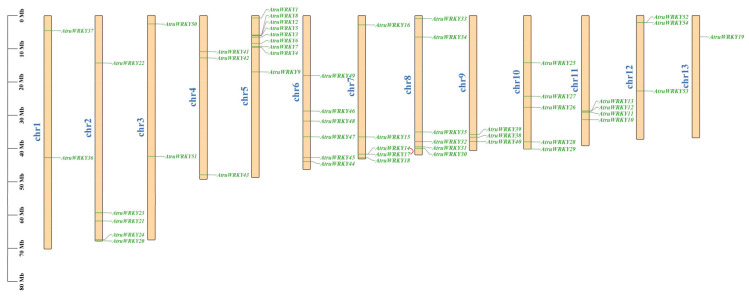
*AtruWRKY* gene distribution across 13 chromosomes of the *A. truncatum* genome. Tandemly duplicated genes are marked with red.

**Figure 4 genes-12-01867-f004:**
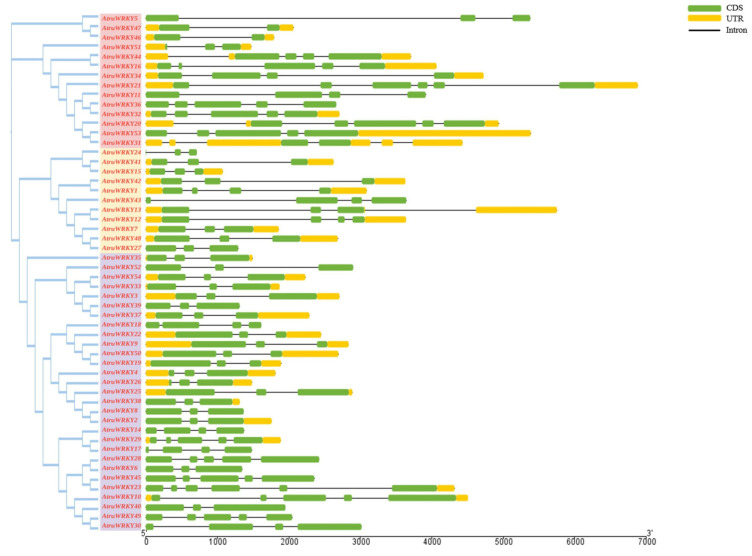
The gene structure of *A. truncatum WRKY* genes according to the phylogenetic relationship. The phylogenetic tree was constructed with the full length sequences of *A. truncatum* WRKY proteins using MEGA6.0, and three groups were marked with different colors. Introns, exons and UTR are represented by black lines, green boxes and yellow boxes respectively.

**Figure 5 genes-12-01867-f005:**
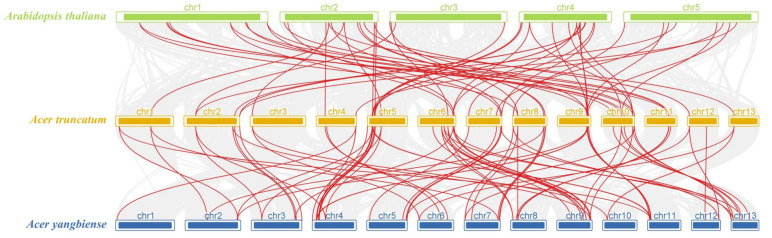
Synteny analyses between the *WRKY* genes of *A. truncatum*. The collinear gene pairs with *AtruWRKY* genes are highlighted in the red lines, while the collinear blocks are marked by gray lines.

**Figure 6 genes-12-01867-f006:**
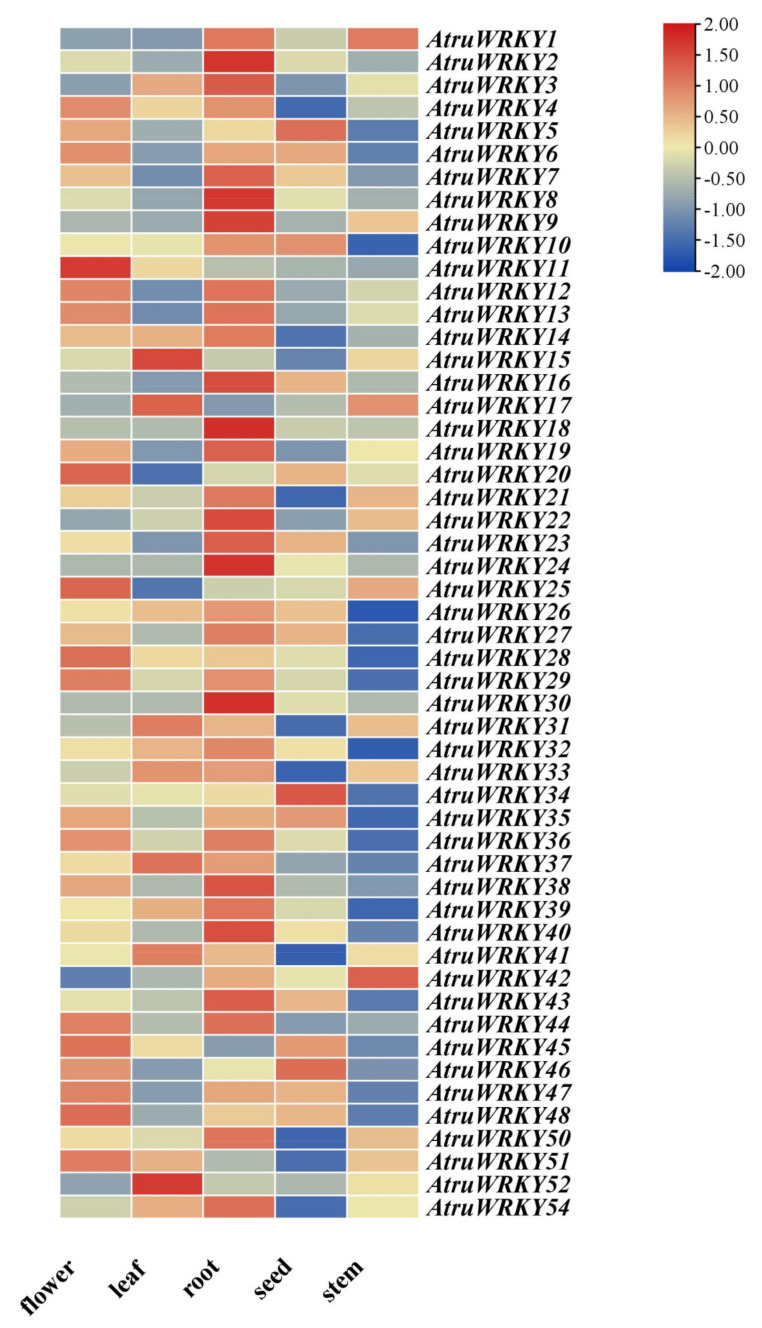
Tissue-specific expression of *WRKY* genes in *A. truncatum*. The color scale shows increasing expression levels from blue to red, which represents log2-transformed FPKM. Gene expression was normalized using Z-scores of fragments per kilobase of exon per million fragments mapped (FPKM) for mean valued.

**Figure 7 genes-12-01867-f007:**
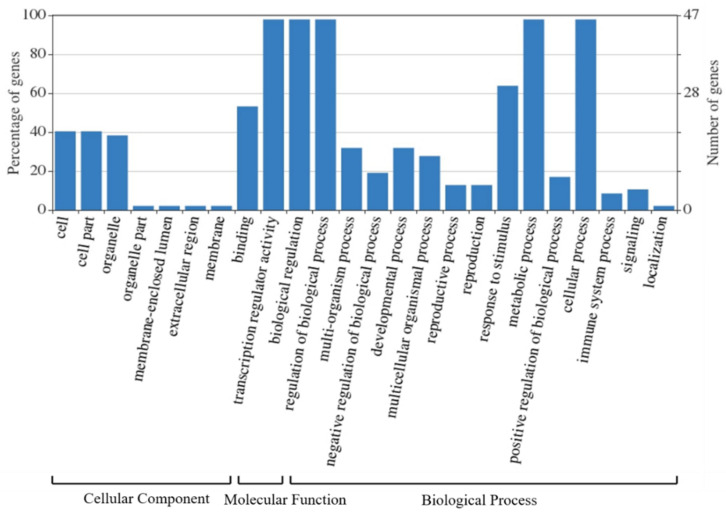
Statistical results for GO annotations of coexpressing genes.

**Figure 8 genes-12-01867-f008:**
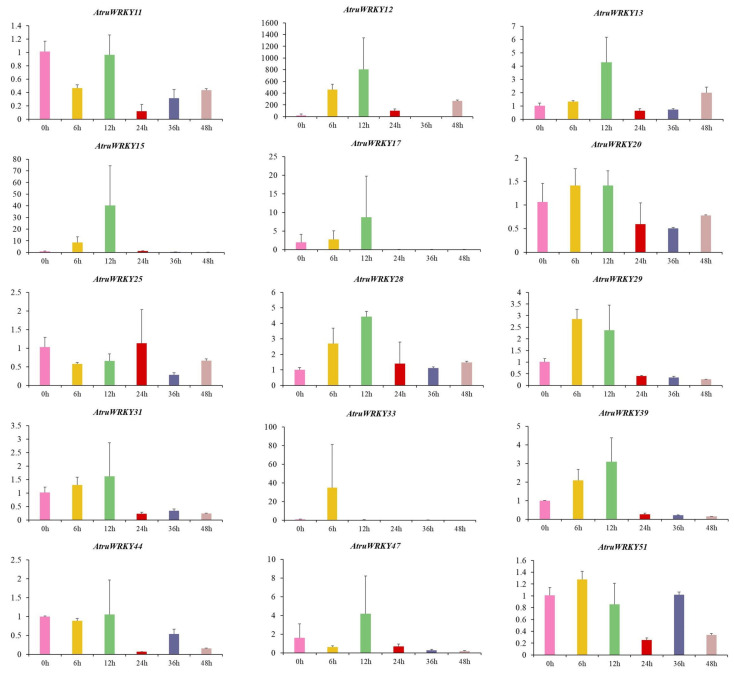
qRT-PCR analysis of *WRKY* genes in *A. truncatum* under cold stress. The Y-axis and X-axis represent the relative expression level and the time course of stress treatment, respectively. Leaves were sampled at 0, 6, 12, 24, 36 and 48 h after 4 °C cold treatments. Data represent the mean ± SD of three technical repetitions.

**Table 1 genes-12-01867-t001:** Information on the WRKY gene family in *A. truncatum*.

Gene Name	Gene ID	SL ^1^	CDS (Length)	AA ^2^	PI ^3^	MW (D) ^4^	Group
*AtruWRKY1*	Atru.chr5.112	cytosol	693	230	7.31	25,946.49	Ⅱ b
*AtruWRKY2*	Atru.chr5.580	nucleus	1122	373	6.13	40,527.27	Ⅲ c
*AtruWRKY3*	Atru.chr5.640	nucleus	1107	368	5.35	40,969.65	Ⅲ a
*AtruWRKY4*	Atru.chr5.983	nucleus	798	265	5.24	30,044.15	Ⅲ c
*AtruWRKY5*	Atru.chr5.585	cytosol	930	309	6.32	33,994.84	Ⅰ a
*AtruWRKY6*	Atru.chr5.856	nucleus	1164	387	8.50	40,844.83	Ⅲ e
*AtruWRKY7*	Atru.chr5.965	nucleus	945	314	6.50	34,739.92	Ⅱ b
*AtruWRKY8*	Atru.chr5.562	nucleus	1122	373	6.13	40,527.27	Ⅲ c
*AtruWRKY9*	Atru.chr5.1597	nucleus	1044	347	9.67	38,515.61	Ⅲ b
*AtruWRKY10*	Atru.chr11.1895	nucleus	1878	625	6.66	68,360.37	Ⅲ e
*AtruWRKY11*	Atru.chr11.1706	nucleus	1551	516	8.85	56,006.74	Ⅰ b
*AtruWRKY12*	Atru.chr11.1674	nucleus	807	268	5.73	29,060.95	Ⅱ b
*AtruWRKY13*	Atru.chr11.1669	nucleus	915	304	6.12	32,948.26	Ⅱ b
*AtruWRKY14*	Atru.chr7.2541	nucleus	1029	342	7.62	38,513.83	Ⅲ d
*AtruWRKY15*	Atru.chr7.1919	nucleus	492	163	5.63	18,559.46	Ⅱ a
*AtruWRKY16*	Atru.chr7.53	nucleus	1482	493	5.62	53,646.88	Ⅰ b
*AtruWRKY17*	Atru.chr7.2542	nucleus	819	272	8.70	30,682.41	Ⅲ d
*AtruWRKY18*	Atru.chr7.2657	nucleus	1020	339	9.33	36,585.20	Ⅲ b
*AtruWRKY19*	Atru.chr13.744	nucleus	1140	379	9.45	41,219.55	Ⅲ b
*AtruWRKY20*	Atru.chr2.3906	nucleus	2241	746	5.78	80,810.51	Ⅰ b
*AtruWRKY21*	Atru.chr2.3616	nucleus	1725	574	6.88	61,806.15	Ⅰ b
*AtruWRKY22*	Atru.chr2.840	nucleus	1086	361	9.84	40,045.18	Ⅲ b
*AtruWRKY23*	Atru.chr2.3488	nucleus	1656	551	7.84	60,264.59	Ⅲ e
*AtruWRKY24*	Atru.chr2.3873	cytosol	216	71	8.95	8012.95	Ⅱ a
*AtruWRKY25*	Atru.chr10.684	nucleus	1548	515	6.05	55,775.07	Ⅲ c
*AtruWRKY26*	Atru.chr10.1574	nucleus	693	230	5.56	25,497.48	Ⅲ c
*AtruWRKY27*	Atru.chr10.1366	nucleus	984	327	6.46	36,108.74	Ⅱ b
*AtruWRKY28*	Atru.chr10.2262	nucleus	1845	614	6.07	65,386.01	Ⅲ e
*AtruWRKY29*	Atru.chr10.2411	nucleus	1038	345	8.57	38,343.75	Ⅲ d
*AtruWRKY30*	Atru.chr8.2579	nucleus	1740	579	5.90	62,385.94	Ⅲ e
*AtruWRKY31*	Atru.chr8.2526	nucleus	831	276	9.56	29,968.99	Ⅰ b
*AtruWRKY32*	Atru.chr8.2350	nucleus	1692	563	6.10	61,892.15	Ⅰ b
*AtruWRKY33*	Atru.chr8.95	nucleus	1050	349	5.50	39,460.85	Ⅲ a
*AtruWRKY34*	Atru.chr8.344	nucleus	1458	485	6.84	53,270.12	Ⅰ b
*AtruWRKY35*	Atru.chr8.1995	nucleus	981	326	4.82	36,337.09	Ⅲ a
*AtruWRKY36*	Atru.chr1.2580	nucleus	1788	595	7.73	65,397.05	Ⅰ b
*AtruWRKY37*	Atru.chr1.252	nucleus	828	275	8.93	30,322.86	Ⅲ a
*AtruWRKY38*	Atru.chr9.2113	nucleus	1011	336	5.41	37,384.57	Ⅲ c
*AtruWRKY39*	Atru.chr9.2017	nucleus	1089	362	5.51	40,377.47	Ⅲ a
*AtruWRKY40*	Atru.chr9.2304	nucleus	1662	553	7.22	59,525.46	Ⅲ e
*AtruWRKY41*	Atru.chr4.410	nucleus	621	206	6.59	22,872.99	Ⅱ a
*AtruWRKY42*	Atru.chr4.480	nucleus	699	232	8.78	26,451.89	Ⅱ b
*AtruWRKY43*	Atru.chr4.3047	nucleus	1296	431	5.80	46,645.38	Ⅱ b
*AtruWRKY44*	Atru.chr6.3354	nucleus	1689	562	6.07	61,083.23	Ⅰ b
*AtruWRKY45*	Atru.chr6.3220	nucleus	1911	636	6.25	69,112.06	Ⅲ e
*AtruWRKY46*	Atru.chr6.1828	nucleus	558	185	9.41	21,199.56	Ⅰ a
*AtruWRKY47*	Atru.chr6.2590	nucleus	612	203	9.32	22,806.53	Ⅰ a
*AtruWRKY48*	Atru.chr6.2076	nucleus	1035	344	6.73	37,984.86	Ⅱ b
*AtruWRKY49*	Atru.chr6.1003	nucleus	1224	407	6.18	45,486.52	Ⅲ e
*AtruWRKY50*	Atru.chr3.267	nucleus	1047	348	9.74	37,932.62	Ⅲ b
*AtruWRKY51*	Atru.chr3.2384	nucleus	435	144	6.43	16,327.83	Ⅰ b
*AtruWRKY52*	Atru.chr12.203	nucleus	1113	370	5.87	39,919.36	Ⅲ a
*AtruWRKY53*	Atru.chr12.1782	nucleus	2307	768	5.31	83,984.96	Ⅰ b
*AtruWRKY54*	Atru.chr12.201_Atru.chr12.202	nucleus	1017	338	5.78	37,079.06	Ⅲ a

Subcellular localization ^1^. The length of the amino acid sequence ^2^. Isoelectric point ^3^. Molecular weight ^4^.

## Data Availability

Genomic data are available in the SRA database of the National Center for Biotechnology Information (NCBI) under the accession number PRJNA557096.

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
