# Peer review of "Genome-Wide Identification and Analysis of the WRKY Gene Family and Cold Stress Response in Acer truncatum"

_genes, 2021, doi:10.3390/genes12121867_

Round 1
Reviewer 1 Report
The authors have performed an in silico characterization of WRKY transcription factors family in the tree species A. truncatum. This family of transcription factors is well studied in model organisms and is known to play a key role in the developmental and stress regulation of genes. Despite having a sequenced genome for A. truncatum, nearly nothing is known about WRKY transcription factors in this species and warrants a thorough investigation of gene annotation before a deeper experimental investigation of the role these transcription factors play. This study attempts to use homology to predict the gene structure, genomic location, and categorize the potential WRKY transcription factors in the A. truncatum. Future studies investigating any specific WRKY transcription factor in this species will benefit from this characterization. Additionally, the authors have preliminarily validated the WRKY transcription factors by showing the sensitivity to cold stress, which is a known role of some WRKY factors in other organisms.
This manuscript is incomplete without a few other analyses. Please see my comments below. Additionally, this reviewer finds missing details about methods. The manuscript will also benefit from re-writing of certain sections and fixing of a lot of errors.
Major
- Re-write: Line 49-76. This entire paragraph is full of examples of what diverse roles WRKY transcription factors play in different plant species. Two to four examples suffice in making the point that WRKY transcription factors are worth investigating. This is not a review article. It is very difficult for readers to follow just examples of the role of transcription factors without a clear goal. The authors must re-write this paragraph in a concise way not exceeding 4-5 statements.
- Missing methods: Section 2.7. Is the qRT-PCR done with oligo d(T) primed reactions, e assessing poly(A)+ mRNAs, or was randomly primed total RNA used for this PCR? The authors must make this clear in the Methods section.
- The authors mention that 94% of the WRKY genes are predicted to have a subcellular localization in the nucleus. Do these WRKY genes have a conserved nuclear localization signal (NLS)?
- The authors suggest that the WRKY genes adjacent to each other may have a tandem duplication relationship between them. If so, then these genes should be part of the same subgroups and be phylogenetically close to each other. Do the authors observe this close relationship? Please discuss.
- How have the authors predicted gene structure? TB tools perform ORF prediction and do gene structure visualization. However, the publication or the tool does not mention performing gene structure prediction. The authors should mention clearly how these predictions were performed and what parameters were used.
- The authors have performed gene structure prediction and used publicly available RNA-seq data to analyze expression. The authors should also use the RNA-seq data to verify their predicted gene structure and either find support for or reject the prediction models. Given the fact that RNA-seq data is available for multiple tissues, the gene structures should find strong support. What percentage of the genes have correctly predicted gene structure? This is a good internal control to bioinformatically verify the quality of these gene structure annotations.
- Line 257 states: 52 out of 54 AtruWRKYs are expressed in all tissue types. This is categorically wrong. As clear from the heatmap there are many of these genes that are not expressed in the stem.
- What is the unit of measurement for the heatmap shown in Figure 6? How can the expression level be negative (blue)? If this is log (fold change), then what tissue type or sample is the expression compared to? The authors must clearly write out the unit.
- How was the order of visualization of AtruWRKY genes in Figure 6 determined? Is it by subgroups? It is very difficult to follow along with the results if the AtruWRKYs are not sorted numerically from 1 through 54? Either the order should be fixed to be numerical, or it should be mentioned how the order was determined. If it is by subgroup, the subgroup type must be noted on the side.
- The authors should mention how the 15 genes were selected for qRT-PCR experiment in figure 8? What criteria were used? Or was it random?
Minor
- The primer sequences for qRT-PCR should be moved to a supplemental table.
- Incomplete sentence: Line 72: “and 12 PmWRKY genes were significantly…”. The authors should complete this sentence.
- There are many writing errors that should be fixed. For example,
- Line 41: The sentence should be “…participate in zinc finger protein interactions”. Delete the extra ‘of’.
- Line 73: There should be a ‘space’ between 11 and EglWRKY.
- Line 89: “..future further..”. One of the redundant words should be deleted.
- Line 108: The wording should be “The A. truncatum genome files..”. The word ‘files’ is missing.
- Sentence structure: Line 37-38. The sentence should be “..transcription factors with a highly conserved protein structure domain[3,4]”, instead of writing ‘conserved’ twice in the long sentence.
- Figure 2 and Figure 3 fonts are extremely small and very difficult to read. The authors must make the font size bigger for clarity.
- Line 223: AtruWRKY2 is repeated twice in the same list. The authors should fix this.
Reviewer 2 Report
The present manuscript corresponds to the identification of WRKY transcriptions factors in A. truncatum. Besides this interesting work, there are few things that should be modified to improve the manuscript.
-
- The present article needs to rephrase some sentences due to grammatical errors or really large sentences (below are indicated some of them).
-
- In Figure 8, the y axis are different for each graph. It is complicated to get some conclusions from the graphs because it is difficult to compare among them. Please change all graphs by having the same y axis, then you can rewrite all your results and conclusions from the new graphs. The levels of expression are different and you can not determine a significant expression level from your graphs.
-
- The 74 known WRKY sequences from A.thaliana were used to identify the orthologous sequence in A.truncatum. How many putative AtruWRKY genes did you get per each A.thaliana gene? Explain better in the text this step, which is crucial for your entire analysis. Provide an explanation about the filters of p-value that you applied. Did you get any duplicated tandem WRKY genes in A.truncatum?
-
Line 23: missing genes, “the results affirmed 54 AtruWRKY genes were divided...”
-
Line 23-26: rephrase the sentence, too long and it generates confusion
-
line 59: misspelling soybean
-
line 60: please write properly the species name of Phytophthora sojae
-
line 60: what does it mean JA? Jasmonic acid? Please write the full name
-
line 61: missing of, “but a lot of studies resported..”
-
line 67 -70: please rephrase the sentence, it is not clear the idea
-
line 72: provide an explanation about the significance of 12 PmWRKY genes, were they upregaultaed or downregulated by cold stress? It is not clear in the sentence.
-
Line 82: any tissue is rich in proteins, it is not meaningful to include it
-
line 108: indicate the genome version and the website from where the documents were donwloaded
-
line 113: missing citation for TBtools, and also indicate the version of the program
-
line 116: NCBI is a huge database with a lot of bioinformatic tools, please explain which tool have you use and the purpose for using it
-
line 119: what does it mean “plant zn clust”?
-
Line 118 – 122: please explain clearly the analysis, the 7 conserved domains were identified in A.thaliana, but then how could you determine it in A.truncatum?
-
Line 125: missing genes, “and isoelectric points (PI) of AtruWRKY genes.”
-
line 126: replace calculated by predicted
-
line 130: please indicate the parameters use to identify exon-intron structure using Tbtools
-
line 148: specify which algorithm has been used to perform the multiple sequence analysis, also indicate the tool used for that purpose (MUSCLE)
-
line 151: explain which species was used as an outgroup for your phylogenetic tree
-
line 153: indicate the genome version and provide the website link
-
line 163: specify the R packages used to standarize and visualize your gene expression data
-
linea 165-166: please provide more information about your samples. From which individuals was obtained the RNA? Was the RNA extracted from leaves? How many grams?
-
Line 172 – 175: more information is required. Indicate all reagents for your qRT-PCR and concentrations. Indicate your negative and positive control of your qRT-PCR.
-
Line 179: replace online alignment by multiple sequence alignment
-
line 231: what do you mean by most complicated? Please provide an explanation
-
line 236: replace “oil palm” by “ A.truncatum
-
line 240: exons are represented by green boxes, not red
-
line 246: the total of AtruWRKY genes are 54 identified. I don't understand how you identified 65 AtruWRKY genes syntenic in A.yangbiense. Please provide an explanation.
-
Line 256-257: all AtruWRKY genes were expressed in all tissues, except 2. From your Figure 6 seems that there are more genes without expression in all tissues tested. Please indicate the 2 genes in the text.
-
Line 269: explain why the functional annotation was only performed for 47 genes, while you have a total of 54 genes identified
-
line 271-272: how many genes have been annotated to take part in membrane structure formation? Please provide the number of genes have been annotated for each GO term
-
line 274-275: I don't understand how you determined the significance in terms of expression level for each biological process. Please explain better the analysis that you have performed and why the GO terms biological processes are significant in expression levels.
-
Line 282: it is no clear in the manuscript why you select 15 AtruWRKY genes for checking the expression levels at different low temperature stages. Please provide an explanation how you select those genes
-
Line 288: AtruWRKY20, AtruWRKY25 and AtruWRKY51 present different expression patterns. AtruWRKY51 is highly expressed the first 12h, it decreases at 24h but then increases again, while the other two genes have different pattern. I don't understand what you want to highligh within your sentence, please rephrase and provide a clear idea.
-
Line 338: what do you mean that all protein sequences contained 2 motifs? Please provide an explanation
-
line 340-343: please rephrase the sentence, it's not understandable the idea behind
-
line 356: it is difficult to distinguish the two genes without differential expression, because looking at Figure 6 seems that all 54 genes present ceratin levels of expression in all tissues. Could you please mention these 2 genes?
-
Line 358: indicate the % of genes that are highly expressed in roots, seeds and stems, to have an easy comparison.
-
Line 365-366: rephrase the sentence
-
line 367-369: rephrase the sentence
-
line 393: add WRKY, “descriptions of the WRKY gene family”
-
line 399: molecular functions and biological processes are general categories, please indicate a more specific GOterms results
Round 2
Reviewer 2 Report
The authors modified their manuscript according to reviewer's suggestions. Thanks to your effort the quality of the article has improved. Besides that, there's still some details that should be modified before being published:
line 180 -183: please provide the p-value and other paramenters (if used) applied to filter out the repetition and blanks.
line 189: please write the full name of plant zinc cluster
line 266: replace “in leaves” to “from leaves”